# Obesity and Breast Cancer: A Case of Inflamed Adipose Tissue

**DOI:** 10.3390/cancers12061686

**Published:** 2020-06-25

**Authors:** Ryan Kolb, Weizhou Zhang

**Affiliations:** 1Department of Pathology, Immunology and Laboratory Medicine, Gainesville, FL 32610, USA; zhangw@ufl.edu; 2University of Florida Health Cancer Center, Gainesville, FL 32610, USA

**Keywords:** obesity, breast cancer, inflammation

## Abstract

Obesity is associated with an increased risk of estrogen receptor-positive breast cancer in postmenopausal women and a worse prognosis for all major breast cancer subtypes regardless of menopausal status. While the link between obesity and the pathogenesis of breast cancer is clear, the molecular mechanism of this association is not completely understood due to the complexity of both obesity and breast cancer. The aim of this review is to highlight the association between obesity and breast cancer and discuss the literature, which indicates that this association is due to chronic adipose tissue inflammation. We will discuss the epidemiological data for the association between breast cancer incidence and progression as well as the potential molecular mechanisms for this association. We will focus on the role of inflammation within the adipose tissue during the pathogenesis of breast cancer. A better understanding of how obesity and adipose tissue inflammation affects the pathogenesis of breast cancer will lead to new strategies to reduce breast cancer risk and improve patient outcomes for obese patients.

## 1. The Statistics and Co-Morbidities of Obesity

Obesity is a condition of having excess body fat that poses a health risk. It is generally defined as having a body mass index (BMI)—an indicator of body fatness determined by dividing the weight of an individual by the square of their height—of ≥30, whereas BMIs from 18.5 to 24.9 are normal weight and BMIs within the range of ≥25 to <30 are considered overweight [1]. Obesity is an increasing public health crisis, as the number of obese adults in the world has approximately doubled since 1980, with an estimated 1.9 billion people or more than 1/3 of the population being overweight or obese [1]. Obesity is a complex multifactorial condition that systemically affects the body and is caused when energy intake is greater than energy expenditure. The increase in the prevalence of obesity over the last 40 years is influenced by many things, including diet, environment, socioeconomic factors, and decreased physical activity due to modern lifestyles [2,3,4].

Obesity is particularly prevalent in western countries, with the American and European regions having the highest prevalence of obesity. In the U.S. and Europe, 28.3% and 22.9% of adults were obese in 2015, a 2.2 and 1.6 fold increase, respectively, since 1980 [1]. This represents both a major public health and economic problem. The public health problem arises from the association of obesity with the increased risk of several chronic health problems, including diabetes, cardiovascular disease, liver disease, and several types of cancer [5,6]. Studies have indicated that every 5 point increase in BMI over 25 results in a 29% increase in overall mortality, a 210% increase in diabetes-associated mortality, and a 41% increase in mortality related to the cardiovascular system [7,8]. Obesity greatly increases the risk of non-alcoholic fatty liver disease (NAFLD)—a chronic liver condition that can lead to nonalcoholic steatohepatitis (NASH) and is one of the most common indicators of transplantation need in Western countries [9]. Histological analyses of liver samples from biopsies, donors, and cadavers have shown that 15% of non-obese people had NAFLD, while 65% of obese individuals and 85% of morbidly obese people (BMI > 40) had NAFLD [10,11,12,13,14]. Obesity is the leading risk factor for the development of type 2 diabetes mellitus [15]. Obesity is associated with both an increased risk and worse outcome for several types of cancer, including liver, gastrointestinal, esophageal, ovarian, and breast cancers [16]. Approximately 3.5% of cancer in men and 9.5% in women are due to obesity, with 14.2% and 19.8% of cancer-related mortality attributed to obesity in men and women, respectively [17,18]. A large meta-analysis by Larsson S.C. and Wolk A., which included more than 7 million people, found that there was a 24% increase in the risk of liver cancer for every 5-point increase in BMI. This study also found that the relative risk of liver cancer is 17% in overweight individuals and 89% in obese individuals [19]. Several studies have shown that obesity, as indicated by BMI, is associated with an increased risk of colon cancer [20,21,22], though other measurements of obesity, such as waist circumference or waist-to-hip ratio, are better predictors of colon cancer risk [23]. A meta-analysis of 19 prospective studies found that a larger waist circumference and waist-to-hip ratio was associated with a relative risk of colon cancer of 1.53 and 1.39, respectively [24]. In melanoma, while the association between obesity and cancer risk is not as clear [25,26,27,28,29,30], some studies have shown that obesity may be associated with disease progression and resistance to therapy [31,32,33,34]. Conversely, some studies have shown that obese patients may respond better to immunotherapies [35,36].

The health risk associated with obesity leads to increased health care costs. This, along with lost productivity due to these chronic health problems, contributes to the economic impact of obesity [37,38]. In 2008, the estimated health care cost associated with obesity in the U.S. was $147 billion, and the annual cost of lost productivity is estimated to be between $3.4 and $6.4 billion [39,40].

## 2. Obesity and Breast Cancer

There have been numerous epidemiological studies and meta-analyses to demonstrate that obesity is associated with both an increased risk of estrogen-receptor (ER)-positive breast cancer in post-menopausal women and worse clinical outcomes independent of menopause status relative to their normal-weight counterparts. Here, we will briefly cover this association, as it has been extensively reviewed [6,41,42]. Several large studies, including the Nurses’ Health Study of more than 87,000 women and the European Prospective Investigation into Cancer (EPIC) study of about 250,000 post-menopausal women, have shown that increased body weight is associated with an increased risk of developing ER-positive breast cancer in postmenopausal women [43,44]. Several meta-analyses and prospective studies have also been performed and support the above conclusion [45,46,47,48]. There are inconsistent results between obesity and breast cancer risk in premenopausal women. In one study, obesity was associated with a decreased risk of developing ER-positive breast cancer in pre-menopausal women, while being associated with a higher risk in post-menopausal women. The authors found no correlation between obesity and ER-negative breast cancer [49]. However, a meta-analysis has indicated that obese pre-menopausal women had a higher risk of developing triple negative breast cancer (TNBC) [50]. This is supported by other studies indicating that obese premenopausal patients are more likely to have TNBC [51,52]. There are also reports showing an association between obesity and the risk of inflammatory breast cancer regardless of menopausal and ER status [53,54,55].

Apart from the increased risk of developing breast cancer, several studies have shown that obesity is associated with worse outcomes for breast cancer patients irrelevant of subtypes. Several meta-analyses have been performed and provide strong evidence for the association between obesity and increased breast cancer mortality. One analysis of 43 studies found a 33% increased rate of death in obese patients [56]. Another meta-analysis of 82 studies found an estimated increased risk of all-cause mortality and breast cancer-specific mortality of 41% and 35%, respectively. This increase in mortality was independent of menopausal and ER status [57]. A large prospective cohort study of breast cancer patients in Denmark showed that obesity was associated with an increased risk of distant recurrence after 5–10 years, with a hazard ratio of 1.42 (95% confidence interval (CI), 1.17 to 1.73) and 1.46 (95% CI, 1.11 to 1.92) for overweight and obese patients [58]. Several other studies have found similar results showing the correlation between obesity and distant breast cancer recurrence [59,60,61,62]. Obesity has also been shown to be associated with an increased risk of developing a secondary malignancy—both at the contralateral breast or at other sites [63].

Obesity also affects other aspects of breast cancer, including diagnosis, treatment, and metastasis. A few studies have shown a correlation between obesity and higher Tumor Node Metastastis (TNM) stage with increased lymph node involvement at time of diagnosis [58,64]. However, this association may be due to diagnostic difficulties and/or reduced adherence to mammographic screening recommendations in obese women [65,66,67]. Obesity is also associated with an increased risk of complications following surgical treatment of breast cancer [42,68,69]. Other studies have shown that obesity is associated with a reduced response to therapeutic treatments, though this may be related to dose-limiting toxicities and undertreatment in obese patients [70,71,72]. However, some studies have suggested a molecular link between obesity and the resistance to these therapies, particularly in relation to adipocytes [73,74,75,76,77]. Obesity has also been linked to increased distal and/or lymph node metastasis [78,79,80,81]. More studies and meta-analyses are still needed to establish the link between obesity and breast cancer progression, response to therapy, and metastasis.

A summary of some of these studies can be found in Table 1.

## 3. Mechanistic Link between Obesity and Breast Cancer

While the epidemiological link between obesity and breast cancer incidence and outcome is well established, the molecular mechanism explaining how obesity promotes breast cancer development and progression is less clear. Various mechanisms have been proposed to explain the correlation between obesity and breast cancer [6,42]. These proposed mechanisms share some common themes, primarily the role of adipocytes and adipose tissue in breast cancer and obesity-associated chronic inflammation.

### 3.1. Adipocytes and Adipokines

Adipocytes are a major component of the breast, ranging from 7–56% of the total volume [82]. It is reasoned that the obesity-associated adiposity or changes that occur in obese adipose tissue affect breast cancer development and progression. Several studies have shown that adipocytes play an important role in breast cancer [82,83,84]. Apart from its role as a depot for energy storage, adipose tissue acts as a secretory organ, producing metabolic substrates (such as free fatty acids and cholesterol), growth factors, hormones, and cytokines referred to as adipokines [85]. Studies have shown that these factors secreted by adipocytes can promote breast cancer initiation, growth, and migration [84,86,87,88]. The co-culture of breast cancer cells with adipocytes increases the proliferation, migration, and invasiveness [89,90,91]. Moreover, the co-culture of adipocytes with breast cancer cells changes the secretome of the adipocytes to be more pro-tumorigenic [91]. In our model of obesity-driven breast cancer in mice, we found that there was an increase in both the size and number of adipocytes within tumors from obese mice compared to lean mice [92]. In humans, tumor infiltration into the surrounding adipose tissue is an indicator of aggressiveness and is associated with poor prognosis [93,94]. These studies suggest that the expansion of mammary adipose tissue during obesity may be an important factor in the association between obesity and breast cancer.

Apart from an expansion of adipose tissue during obesity, obesity modifies the behavior and secretome of adipose tissue. Studies have discovered that obese postmenopausal women have an increased expression of aromatase in adipocytes, which leads to increased circulating levels of estrogens. This increase in circulating estrogens may contribute to the risk of ER-positive breast cancer in obese postmenopausal women [95]. However, counterintuitively, several studies have shown that circulating estrogens have a protective role against the development of breast cancer in obese women [96,97].

Obesity leads to an increase in the adipokine leptin and a decrease in adiponectin. Numerous studies have been conducted on the role of these adipokines in breast cancer. A meta-analysis of 23 studies found that high leptin levels are positively correlated with the increased risk of breast cancer [98]. Other studies have indicated that a high leptin level is an important indicator of breast cancer risk in obese postmenopausal women [99,100]. Moreover, numerous studies have shown that leptin is pro-tumorigenic, with the capability to promote the proliferation, transformation, and survival of mammary epithelial and breast cancer cells [42,101]. In contrast, meta-analyses have indicated that there is a correlation between low adiponectin levels and breast cancer risk in postmenopausal women [102,103]. Both in vitro and in vivo studies have shown that adiponectin can inhibit the proliferation and survival of breast cancer cells [104,105]. Circulating levels of other adipokines have also been associated with breast cancer. Resistin and visfatin, two adipokines that are elevated in obese adipose tissue, have been positively correlated with breast cancer risk, poor prognosis, tumor size, TNM stage, lymph node involvement, and metastasis [106,107,108,109,110,111,112].

### 3.2. Adipose Tissue Inflammation

Inflammation is a common factor between many obesity-associated comorbidities, including cancer, suggesting that obesity-associated inflammation may be a common factor in the pathophysiology of several obesity-related health risks [6,113,114,115,116]. Obesity leads to low-grade chronic inflammation that affects multiple organs in the body, including the white adipose tissue, liver, intestines, pancreas, muscle, and nervous system [117,118,119,120,121,122]. Studies have shown that obesity-associated inflammation starts with excess energy intake (overeating), which leads to an energy imbalance and the activation of various signaling pathways in metabolic tissues, including the white adipose tissue, liver, and muscle. This results in the upregulation of inflammatory mediators and inflammatory cytokines from both the tissue resident cells (adipocytes, hepatocytes, etc.) as well as from immune cells within those tissues [123,124]. Because we are discussing breast cancer, we will focus on adipose tissue inflammation and its association with breast cancer.

Excess energy is stored as triglycerides in adipocytes, which—if it is too much—leads to the expansion of adipocytes via both hypertrophy, an increase in size, and hyperplasia, an increase in number [125]. However, studies have shown that adults have a finite number of adipose stem cells, thus limiting adipocyte hyperplasia. This suggests that adipose tissue expansion is primarily due to adipocyte hypertrophy. Moreover, adipocyte hyperplasia may be protective against obesity-associated inflammation, indicating that adipocyte hypertrophy plays a larger role in obesity-associated inflammation [125,126,127]. When adipocytes undergo hypertrophy, their secretome changes to a more inflammatory one due to mechanical stress from expansion and decreased oxygen supply. As adipocytes expand in size, they experience mechanical stress from interactions with neighboring cells and the extracellular matrix. This leads to endoplasmic reticulum (ER) stress and the activation of the unfolded protein response (UPR), a process that has been shown to enhance the expression and secretion of inflammatory cytokines, including interleukin (IL)-6 and tumor necrosis factor (TNF) α [128,129,130]. Adipose tissue expansion also leads to increased hypoxia, which leads to increased levels of chemokines and inflammatory cytokines [131,132]. Adipocyte hypertrophy also leads to adipocyte cell death, which causes the release of danger-associated molecular patterns (DAMPs), chemoattractants, and inflammatory cytokines, resulting in the recruitment and polarization of macrophages to a more inflammatory state. Macrophages surround dying or dead adipocytes and form a histologically distinct feature called a crown-like structure (CLS) [133,134]. Macrophages in obese adipose tissue secrete inflammatory cytokines, such as IL-1β, IL-6, and TNFα, that contribute to both local and systemic inflammation [133]. The dysregulation of adipokines during obesity also contributes to adipose tissue inflammation. Leptin has been shown to modulate immune cells, including macrophages, to induce the secretion of inflammatory cytokines [135]. Conversely, adiponectin has been shown to have anti-inflammatory properties [136]. In summary, obesity causes adipocyte hypertrophy, resulting in increased leptin, ER stress, hypoxia, and/or adipocyte cell death, which can lead to the secretion of inflammatory cytokines as well as the recruitment and polarization of inflammatory macrophages (Figure 1).

There is strong evidence that obesity-associated inflammation is driving both the increased risk of breast cancer and poor breast cancer prognosis in obese women. In obese postmenopausal women, there is a positive correlation between circulating C-reactive protein (CRP)—a marker of inflammation—and breast cancer risk [137]. CRP levels are also associated with shorter disease-free survival [138]. Transcriptomic analysis of RNA sequencing data from ER-positive breast cancer patients has found that genes associated with inflammation and immune cell trafficking are enriched in obese patients [139]. The other proposed mechanistic links between obesity and breast cancer are associated with inflammation as well. Inflammation has been shown to lead to increased aromatase expression in adipocytes [140,141]. Leptin and resistin have been shown to induce inflammatory signaling in various immune cells [135,142,143], while adiponectin has anti-inflammatory properties [136]. Obesity-associated inflammation also promotes the development of metabolic syndrome and insulin resistance [144]. Insulin resistance leads to increased levels of insulin and its associated growth factors, in particular insulin-like growth factor 1 (IGF-1), and the development of type 2 diabetes [145]. Type 2 diabetes, insulin resistance, and high levels of IGF-1 have been linked to the increased risk of breast cancer [87,146,147,148]. Moreover, IGF-1 and insulin levels are positively correlated with recurrence and mortality. Other studies have shown that IGF-1 signaling is associated with a poor prognosis [149,150].

Many inflammatory cytokines that are upregulated in inflamed adipose tissue under the obese condition are associated with breast cancer risk and progression. TNFα, IL-6, and IL-1β are all elevated in obese breast cancer patients and are associated with breast cancer progression and outcomes [151,152,153,154]. However, direct evidence linking these inflammatory cytokines to obesity-associated breast cancer risk and outcomes is limited. Using syngeneic breast cancer models in mice, we have shown that obesity leads to the NLRC4-inflammasome-mediated activation of IL-1β in infiltrating macrophages, which in turn promotes increased angiogenesis and disease progression. The loss of the NLRC4 inflammasome or inhibition of IL-1β is sufficient to abolish the increased tumor progression in obese mice, providing strong evidence that obesity-associated inflammation drives breast cancer progression [155]. Further studies found that IL-1β promoted obesity-driven breast cancer angiogenesis and progression via the upregulation of angiopoietin-like 4 (ANGPTL4) in adipocytes [92]. Other studies have also linked obesity-associated inflammation and angiogenesis to breast cancer [83,156,157]. These studies provide strong evidence that obesity-associated chronic inflammation in adipose tissue is one of the main driving factors of the association between obesity and breast cancer risk and poor outcomes.

A summary of the proposed mechanistic links between obesity and breast cancer are presented in Figure 2.

## 4. Adipose Tissue Inflammation and Breast Cancer in Non-Obese Women

The evidence suggests that obesity-associated chronic inflammation in adipose tissue is, at least in part, driving the correlation between obesity and breast cancer. Given that adipocytes are an important component of the breast cancer tumor microenvironment [82], it is reasonable to speculate whether obesity-associated inflammation can promote breast cancer in non-obese women, or if it is specific to obesity. The idea that adipose tissue inflammation may promote obesity-related co-morbid conditions independently of obesity is supported by studies showing that adipose tissue inflammation is correlated with insulin resistance, metabolic syndrome, and type 2 diabetes in normal weight individuals [158,159,160].

In order to study the association between adipose tissue inflammation and breast cancer, markers for inflamed adipose tissue need to be identified. As stated previously, obesity leads to adipocyte hypertrophy, cell death, the recruitment of macrophages, and the formation of CLS (Figure 1). The appearance of CLS in adipose tissue is considered a histological marker of inflammation. Indeed, studies have shown that the abundance of CLS is elevated in the adipose tissue of obese patients and animals and is an indication of the level of macrophage infiltration, inflammatory cytokines, and aromatase expression [134,140,161]. Moreover, the abundance of CLS in adipose tissue is associated with insulin levels, insulin resistance, and type 2 diabetes [162]. However, while CLSs are more prevalent in obese patients, they do occur in non-obese patients. One small study found that while CLSs are found in 75% of obese patients, they are also found in 8% of normal weight patients [140]. This is supported by other studies finding similar rates of CLS in adipose tissue in obese and non-obese patients [159,163].

There are a few studies showing that the presence of CLS in mammary adipose tissue may increase the risk of breast cancer independently of obesity. One study looked for CLS in mammary adipose tissue from benign biopsies and found that a high CLS count was associated with an increased risk of developing breast cancer after adjusting for BMI, indicating that a high CLS count may be an independent marker of breast cancer risk [164]. In another study, Iyengar, NM. et al. evaluated tissues from women undergoing mastectomy and found that CLSs are associated with elevated insulin levels, CRP, and IL-6. Furthermore, a high CLS count is associated with a shorter distant relapse-free survival (HR 1.83; 95% CI, 1.07–3.13) after adjusting for BMI [165]. In a similar study, Iyengar, NM. et al. looked at samples from 72 normal weight women undergoing mastectomy for breast cancer risk reduction and found that CLS abundance is correlated with higher levels of aromatase expression, leptin, insulin, and CRP [159]. These studies support that adipose tissue inflammation, as evidenced by the presence of CLS, may increase the risk of breast cancer and promote breast cancer progression independently of obesity.

## 5. Conclusions

There is sufficient evidence to support the idea that obesity is a risk factor for the development of breast cancer and is associated with increased mortality. The association between obesity and breast cancer outcomes may be because obese patients tend to have a higher TNM stage when diagnosed, though some studies have shown that the correlation between obesity and mortality is independent of stage [58,64,80]. This correlation between obesity and breast cancer mortality warrants the development of better strategies for treating obese patients. Some of the correlation with poor outcome may be related to the suboptimal dosing of chemotherapies in obese patients due to dose-limiting toxicities. New guidelines recommend using the full dose of chemotherapy based on actual body weight in obese patients, because studies have shown that this does not increase treatment-related toxicity [166,167]. Other studies have looked at the use of metformin in treating obese patients as it has been shown to reduce the risk of several cancers, including breast cancer [168,169,170]. Interestingly, some studies have shown that obese patients may respond better to immunotherapies [35,36].

Given the evidence that adipose tissue and in particular adipose tissue inflammation is a major driver of the association between obesity and breast cancer, targeting adipocyte-secreted factors may be a viable therapeutic approach for improving outcomes for obese patients. Several agents targeting the chemokines and cytokines secreted by adipocytes which are elevated by obesity have been used in clinical and preclinical studies. These include Cenicriviroc, an inhibitor of C-C motif Chemokine Receptor 2 (CCR2), the receptor for C-C motif Chemokine Ligand 2 (CCL2), which is important in the recruitment of macrophages to obese adipose tissue [171,172]; Tocilizumab, an IL-6 receptor antibody [173]; Canakinumab, an IL-1β antibody [174]; and Infliximab, a TNFα antibody [175]. We recently showed that obesity-associated inflammation promotes the upregulation of ANGPTL4 in adipocytes, which drives breast cancer angiogenesis and progression. As such, we have developed antibodies against human ANGPTL4 that inhibit ANGPTL4-induced angiogenesis [92]. Therapies which target adipocyte-secreted factors warrant further investigation for treating obese breast cancer patients.

Another possibility is to treat the obesity itself, and there are a few clinical studies looking at the effect of weight loss on breast cancer, including the Breast Cancer WEight Loss (BWEL) study (NCT0270826). The effect of weight loss as a method for lowering breast cancer risk is supported by a few studies, including a large prospective study that found that sustained weight loss in women over 50 is associated with a reduced risk of developing breast cancer [176]. Other studies looking at obese patients who underwent bariatric surgery for weight loss have shown that they have a reduced risk of several types of cancer, including breast cancer [177,178]. Apart from potentially reducing the risk and/or progression of breast cancer, several studies have shown that weight loss can improve the quality of life of breast cancer patients [179,180]. While more studies need to be performed to further support these findings, weight loss intervention may be beneficial for obese breast cancer patients.

Many studies support that obesity-associated inflammation—particularly in mammary adipose tissue—is a major contributor to this association. Interestingly, some studies suggest that mammary adipose tissue inflammation may be a risk factor for breast cancer in normal weight individuals [159,164,165]. There are people that fall into a category referred to as metabolically obese with a normal weight. Though having a BMI in the normal range (18.5–24.9), these individuals tend to have higher adiposity, increased adipose tissue inflammation, and are at a higher risk of developing metabolic syndrome and cardiovascular diseases [158,160,181,182]. It is possible that metabolically obese but normal weight people are also at a higher risk of developing obesity-related cancers including breast cancer, though more studies need to be conducted to support this connection. As obesity and many of the inflammatory cytokines upregulated in obese adipose tissue are positively correlated with breast cancer mortality [152,153], studies should be performed to see if adipose tissue inflammation is a prognostic indicator in both obese and normal weight patients. In summary, an increasing body of evidence strongly supports that obesity-associated inflammation in mammary adipose tissue drives the correlation between obesity and breast cancer risk and poor outcomes. Furthermore, adipose tissue inflammation may promote breast cancer independently of obesity.

## Figures and Tables

**Figure 1 cancers-12-01686-f001:**
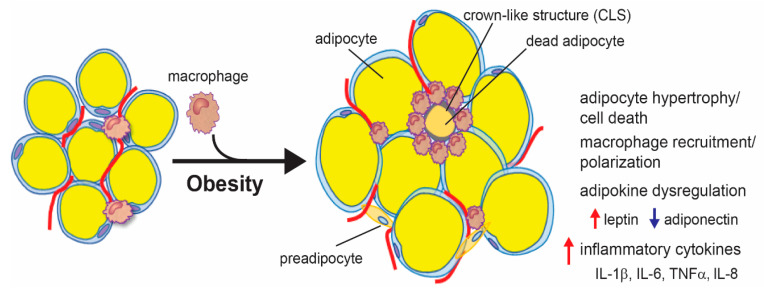
Obesity-associated adipose tissue inflammation. Excess energy intake during obesity leads to adipocyte hypertrophy and cell death. This leads to the secretion of inflammatory cytokines and chemokines, which induce the recruitment and polarization of macrophages. Macrophages surround dead adipocytes, forming a crown-like structure (CLS) and secreting inflammatory cytokines. Adipocyte hypertrophy also leads to an increase in leptin and a decrease in adiponectin, which further induces the expression of inflammatory cytokines.

**Figure 2 cancers-12-01686-f002:**
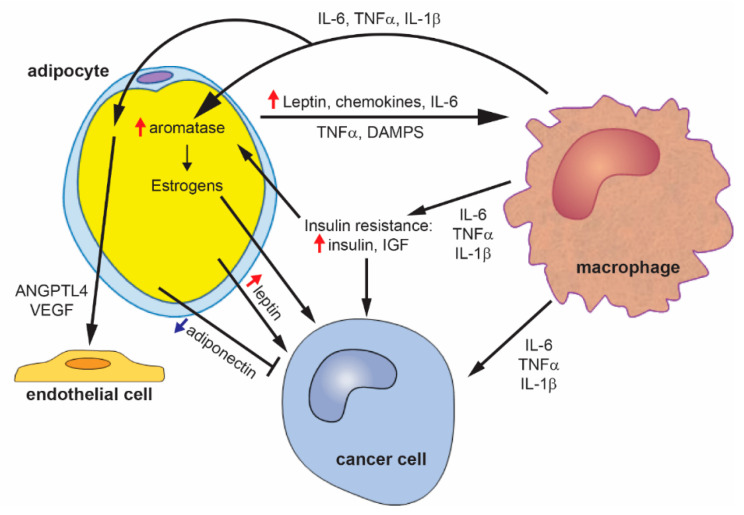
Mechanisms of obesity-driven breast cancer. Obesity leads to adipocyte hypertrophy, which induces the secretion of inflammatory cytokines, chemokines, and leptin. These adipokines then induce the recruitment and polarization of macrophages. Macrophages secrete inflammatory cytokines, which can act directly to promote breast cancer, act on adipocytes to increase the expression of aromatase and estrogen production, and induce the expression of pro-angiogenic factors. Adipose tissue inflammation also promotes the development of insulin resistance, leading to increased insulin and insulin-like growth factor (IGF). Insulin and IGF can directly promote breast cancer. Leptin from adipocytes can also directly act on cancer cells. Obesity leads to a decrease in adiponectin, which inhibits breast cancer.

**Table 1 cancers-12-01686-t001:** Epidemiological studies of obesity and breast cancer. Abbreviations: RR, relative risk; HR, hazard ratio; OR, odds ratio; BC, breast cancer; CI, confidence interval.

Study Type	Cohort	Findings	Reference
Breast Cancer Risk			
Prospective	87,143 postmenopausal women	RR 1.45 (95% CI 1.27–1.66) of ER + BC in women who gained 25 kg or more since age 18	[36]
Prospective	242,918 postmenopausal women	HR 1.23 (95% CI 1.15–1.35) for ER + BC in women with a BMI > 29	[37]
Meta-analysis	57 studies from 1985–2011	RR 1.25 (95% CI 1.07–1.46) for postmenopausal breast cancer in obese women	[38]
Meta-analysis	89 studies from 1980–2012	RR 1.39 (95% CI 1.14–1.70) for ER + BC in obese postmenopausal women No association between BMI and ER- BC	[39]
Prospective	67,142 postmenopausal women	HR 1.86 (95% CI 1.60–2.17) ER + BC in women with BMI > 35.0 No association with ER- BC	[41]
Meta-analysis	31 studies from 1970–2007	33% increase risk of ER + BC for every 5 point increase in BMI in postmenopausal women 20% decrease risk of ER + BC in obese premenopausal women	[42]
Meta-analysis	11 studies through 2012	OR 1.43 (95% CI 1.23–1.65) for TNBC in obese premenopausal women	[43]
Prospective	620 patients	Increase risk of all subtypes of inflammatory breast cancer in women with BMI > 25	[46]
Outcomes			
Meta-analysis	43 studies from 1965–2005	HR 1.33 for overall (95% CI 1.21–1.47) and BC-specific survival (95% CI 1.19–1.50)	[49]
Meta-analysis	82 studies through 2013	RR 1.41 (95% CI 1.29–1.53) for overall mortality for obese women RR higher in premenopausal women (RR 1.75) then postmenopausal women (RR 1.34)	[50]
Prospective	18,967 women with early stage BC	HR 1.46 for developing distant metastasis after 10 years in obese women	[51]
Meta-analysis	26 studies through 2012	RR 1.37 (95% CI 1.20–1.57) of contralateral BC in obese women	[56]

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
