# Peer review of "Obesity and Breast Cancer: A Case of Inflamed Adipose Tissue"

_cancers, 2020, doi:10.3390/cancers12061686_

Round 1
Reviewer 1 Report
- the review is not weel organized.
- the paragraphs on liver and intestinal inflammation are not useful for the review aim which is to explain the link between obesity and breast cancer
- does the review report data from epidemiological, in vivo, in vitro experiemts? the authors should be more precise about the reported evidence.
- the authors report dtaa about leptin. WHat about adiponectin or resistin or visfatin?
- A specific role for the infiltrated adipose tissue has been reported in breast cancer. the authors should deepen this point
- references are too dated.
- none of the 2 figures is specific for breast cancer and does explain the link between adipose tissue and breast.
- overall the manuscript is confusing, and important resent data are missing.
Reviewer 2 Report
This review by Kolb and Zhang described the relationship between and breast cancer, with a particular focus on potential molecular mechanisms involved.
In my opinion, this work is well organized and well written, using appropriate and relevant references. The Authors first describe evidence from epidemiological study, and then summarize potential mechanisms suggested by experimental studies.
I have only minor comments. For instance, it could be helpful for readers a table/figure summarizing potential mechanisms underpinning the relationship. Moreover, it would be interesting to make a discussion on the effect of weight loss program on the quality of life of breast cancer survivors. Indeed, high prevalence of this cancer and improvement in patient management have increased the proportion of survivors. Please consider the following article https://doi.org/10.3390/cancers12020322.
Reviewer 3 Report
In this manuscript Kolb et al, describe the significance and role of obesity in breast cancer a case of inflamed adipose tissue. Specifically, authors try to explore the recent studies findings to discuss the epidemiological data for the association between breast cancer incidence and progression as well as the potential molecular mechanism for how obesity promotes breast cancer incidence and progression. In this study, authors are focused on the role inflammation within the obesity-associated adipose tissue during pathogenesis of breast cancer. In this article authors collected vast information from review literature and very nicely incorporated in the manuscript.
Overall this is an interesting review article with great potential. Collectively, this study conducted meticulously and appears to be well designed and the collected information support the conclusion. The review literature are clearly presented, and appears solid and their analyses are reasonable. Simultaneously, I have some minor suggestions need to be addressed by authors.
Comments:
- In the introduction section need some literature about how obesity plays an important role in other cancer such as melanoma in preclinical models by citing these papers in addition with others. (See, Malvi et al, Elevated circulatory levels of leptin and resistin impairs therapeutic efficacy of dacarbazine in melanoma under obese state. Cancer Metabolism. 2018 Feb 7;4:21. eCollection 2016. Malvi et al, Weight control interventions improve therapeutic efficacy of dacarbazine in melanoma by reversing obesity-induced drug resistance. Cancer Metabolism. 2016 Dec 7;4:21. eCollection 2016. Malvi et al, Controlling obesity reduces rapid progression of melanoma: Role of adipokines. Molecular Oncology 2014, S1574-7891(14)00282-8).
- In the introduction section need some literature about how different strategies are used to treat breast cancer in preclinical models by citing these papers in addition with others (See, Mohammad et al., Strategy to enhance the efficacy of doxorubicin in breast and hepatocellular carcinoma cells by methyl- β-cyclodextrin: Activation of p53 and involvement of Fas receptor ligand. Scientific Reports 2015 7;5:11853. doi: 10.1038/srep11853, Muhammad et al, Bitter melon extract inhibits breast cancer growth in preclinical model by inducing autophagic cell death. Oncotarget. 2017 Aug 03. doi: 10.18632/oncotarget.19887, Muhammad et al, Anti-miR-203 suppresses breast cancer growth and stemness by targeting SOCS3. Oncotarget. 2016 Aug 10. doi: 10.18632/oncotarget.11193)
- Can the authors make the more concrete figure of signalling pathway and flow chart of pathway that affected in breast cancer in relation with obesity? There is no figure of signalling pathway at the moment, thus it might be difficult for readers to understand the contents.
- Authors have not done extensive revision of manuscript in terms of the language and sentence forming. There are several syntax errors and language of the manuscript should be edited. English language needs extensive corrections as this is undermining the merits of the manuscript. Please work on the manuscript writing part. There are so many places where unnecessary words have been used.
- There are many places authors could slightly modify the sentences and no need to write long sentences.
- The authors need to rethink about how to present their view and opinions on this topics. A review is to present your scientific conclusion or hypothesis after analyzing current available data, not just pile up existing data and trials. Wish the authors first ask ourselves what you intend to tell the audience then organize a more concentrated, short, but hitting the point review.
- Please try to correct typological errors throughout the manuscript. I found many typological errors at various places.
Round 2
Reviewer 1 Report
- the aim of the review is not stated
- the first paragraph title is "1. The statistics and co-morbidities of obesity", but in the paragraph the authors only report few data about cancer
- some concepts are repetitive (i.e. the deinition of obesity)
- it would be useful if the authors add info about clinical and therapeutic approaches involving adipose tissue-related factors.
